# Is Frailty Associated with Worse Outcomes After Major Liver Surgery? An Observational Case–Control Study

**DOI:** 10.3390/diagnostics15050512

**Published:** 2025-02-20

**Authors:** Sorinel Lunca, Stefan Morarasu, Andreea Antonina Ivanov, Cillian Clancy, Luke O’Brien, Raluca Zaharia, Ana Maria Musina, Cristian Ene Roata, Gabriel Mihail Dimofte

**Affiliations:** 12nd Department of Surgical Oncology, Regional Institute of Oncology (IRO), 700483 Iasi, Romania; sdlunca@yahoo.com (S.L.); i_antonina@yahoo.com (A.A.I.); raluca.zaharia11@yahoo.com (R.Z.); musina.anamaria@gmail.com (A.M.M.); roatacristianene@gmail.com (C.E.R.); gdimofte@gmail.com (G.M.D.); 2Department of Surgery, Grigore T. Popa University of Medicine and Pharmacy, 700115 Iasi, Romania; 3Department of Colorectal Surgery, Tallaght University Hospital, D24 NR0A Dublin, Ireland; clancyci@tcd.ie (C.C.); lukemobrien13@gmail.com (L.O.); 4Trinity College, University of Dublin, D02 PN40 Dublin, Ireland

**Keywords:** frailty, liver, cancer, surgery, oncology, morbidity, complications

## Abstract

**Background**: The rate of morbidity after liver surgery is estimated at 30% and can be even higher when considering higher-risk subgroups of patients. Frailty is believed to better predict surgical outcomes by showcasing the patient’s ability to withstand major surgical stress and selecting frail ones. **Methods**: This is a single-centre, observational case–control study on patients diagnosed with liver malignancies who underwent liver resections between 2013 and 2024. The five-item modified Frailty Index (mFI-5) was used to split patients into frail and non-frail. The two groups were compared in terms of preoperative, operative and postoperative outcomes using a chi-squared and logistic regression model. **Results**: A total of 230 patients were included and split into two groups: non-frail, NF, *n* = 90, and frail patients, F, *n* = 140. Overall, F patients had a higher rate of morbidity (*p* = 0.04) but with similar mortality and length of stay. When considering only major liver resections, F patients had a higher probability of posthepatectomy liver failure (LR 6.793, *p* = 0.009), postoperative bleeding (LR 9.541, *p* = 0.002) and longer ICU stay (LR 8.666, *p* = 0.003), with similar rates of bile leak, surgical site infections, length of stay and mortality. **Conclusions**: Frailty seems to be a solid predictor of posthepatectomy liver failure in patients undergoing major liver resections and is associated with a longer ICU stay. However, mortality and surgical morbidity seem to be comparable between frail and non-frail patients.

## 1. Introduction

Surgery is the mainstay of treatment in both primary and secondary liver malignancies. Once deemed operable, almost all patients are candidates for upfront surgery as it is proven to improve overall and disease-free survival alongside systemic therapy. One factor to consider is that oncological liver resections are one of the most complex abdominal procedures surgeons face in their practice, and they are usually performed on difficult, malnourished patients, often with pre-existing chronic liver diseases and previous abdominal surgeries (i.e., patients with liver metastases) [1,2,3,4].

The rate of morbidity after liver surgery is estimated at 30% and can be even higher when considering higher-risk subgroups of patients [5,6]. Now, more than ever, elderly patients are diagnosed and treated for primary and secondary liver cancer, and, with the increasing life expectancy, major surgery for geriatric patients will be indicated more frequently. After major surgery, one in seven elderly patients die in the first year, and this must be considered when planning major interventions [7]. Age alone is not enough to stratify at-risk patients, but frailty may be a solid adjunct. Frailty is a structured syndrome characterised by a decreased potential to handle physiological stress. It is a clinical entity capable of selecting patients who may have more difficulties in maintaining homeostasis after stress, including surgical trauma [8,9,10]. While frailty is driven by aging, it is not always present in elderly patients.

In clinical practice, frailty can be defined and quantified through various scores. The 11-item modified frailty index (mFI-11) is one of the most used scores and has several publications to back up its validity. A more concise alternative to the 11-item assessment is the 5-item modified frailty index (mFI-5), which has shown similar results when compared to the former [11,12]. In this study, we aimed to assess the ability of the mFI-5 to predict postoperative complications in patients undergoing oncological liver resections, both minor and major, in a case–control design.

## 2. Materials and Methods

### 2.1. Design and Setting

This is a single-centre, single-department, single-surgeon, observational case–control study on patients diagnosed with liver malignancies, both primary and secondary, who underwent surgery at our institution between 2013 and 2024. All patients underwent standard oncological work-up and management based on multidisciplinary meetings. All patients were treated and followed at our institution.

### 2.2. Inclusion and Exclusion Criteria

The STROBE checklist [13] was adhered to (Figure 1). All patients with primary or secondary liver tumours were included. Only patients who underwent minor (less than three segments) or major (more than three segments) liver resection were included. Biopsies and intraoperative ultrasounds only were excluded.

### 2.3. Data Analysis

Data were collected from a prospectively maintained database on liver resections including perioperative, short-term and long-term outcomes data. The mFI-5 was calculated based on the presence of 5 pre-existing conditions: congestive heart failure, diabetes, chronic obstructive pulmonary disease, functional status and high blood pressure. The mFI-5 was applied to split patients into two subgroups: non-frail (NF) patients, with an mFI-5 score of 0, and frail patients (F), with an mFI-5 score of 1 or more. Data on demographics (age, sex, race, BMI, smoking status, chronic medication, immunosuppressant or steroid use and neoadjuvant therapy); comorbidities (including hepatic, cardiac, respiratory, metabolic, renal or other); diagnosis; the type of operation; the description of the procedure; intraoperative events; the duration of surgery; blood loss; pre- and postoperative blood; general, medical and surgical morbidity; and in-hospital and 30-day mortality were extracted. Fisher’s exact test for qualitative variables was used instead of the chi-square test when the number of datasets and theoretical counts was less than five. A *T*-test (pooled variance) was used for quantitative variables to compare means. A *p* value < 0.05 (95% CI) was considered significant. A binary logistic regression model was used to assess the probability of a patient being non-frail vs. frail given the dataset of patient characteristics and short-term outcomes. A similar design was used to assess the probability of postoperative outcomes (e.g., morbidity, bile leak, liver failure). A goodness-of-fit (GoF) test was used to establish the quality of the model, considering that the assumption brings a significant amount of information if the likelihood ratio (LR) for chi-squared is less than 0.05. Type II analysis was used to interpret the contribution of each predictor variable to the outcome variable. Confusion plots were used to visualise the relationship between predictions and actual values in a synthetic manner. The Goodness of Classification Index (GCI) was calculated to rate the performance of the statistical model. For significant variables of interest, ROC curves were determined to establish the sensitivity and specificity compared to a control random model, where an AUC of 0.7–0.9 was considered good. An AUC of 0.9 was considered excellent. Data analysis was performed using XLSTAT software version 2024.3 [14,15,16].

## 3. Results

### 3.1. Patients’ Characteristics

A total of 230 patients were included in the study (Figure 1) and split into two groups: non-frail, NF (mFI = 0), *n* = 90, and frail patients, F (mFI ≥ 1), *n* = 140. The mean age was 57.3 vs. 66.4 years old, with F patients being significantly older (*p* < 0.0001) with a mean difference of 1.04 years. There were 55.5% males (*n* = 50) and 63.5% females (*n* = 89). Body mass index (BMI) was significantly higher in the F group (*p* = 0.0001), with a mean difference of 0.53. Most patients in the F group scored 1 (53.5%, *n* = 75), with only 2 patients scoring an mFI of 4. As expected, there were significantly more patients with comorbidities in the F group: viral hepatitis (*p* = 0.0001), cirrhosis (*p* = 0.0001), diabetes (*p* < 0.00001), chronic heart failure (CHF, *p* < 0.00001) and chronic kidney disease (CKD, *p* = 0.0001). Also, the F group had significantly more ASA 3 patients (*n* = 53 in NF vs. *n* = 102 in F, *p* = 0.031). In terms of preoperative blood, only platelets (PLT, 239 × 10^3^ in NF vs. 201 × 10^3^ in F) were significantly lower in the F group (*p* = 0.003) which is expected given the higher number of patients with liver disease in the latter. In terms of operative factors, there were similar rates of major liver resections in both groups (*p* = 0.089) and similar rates of intraoperative blood loss (*p* = 0.272); however, operative time was significantly longer in the NF group (*p* = 0.004), with a mean difference of 39 min (Table 1).

### 3.2. Postoperative Outcomes

Overall, there was a higher rate of morbidity in the F group (*p* = 0.04), largely attributed to a higher rate of hospital-acquired infections (*p* = 0.01), such as urinary tract infections, central line infections or Clostridium difficile infections. However, the rate of surgical or medical complications was similar between the two groups. There was no difference in terms of the length of stay (12.8 vs. 13.5 days), length of ICU stay (2.7 vs. 3.3), reintervention rate or 30-day mortality (Table 2).

### 3.3. Major Liver Resections

Patients who underwent major liver resections (i.e., three or more segments) were analysed further as a subgroup. A logistic regression model was used to assess the probability of a patient being classified as NF or F based on their postoperative outcomes (i.e., null hypothesis). The GoF test and test of null hypothesis showed that the probability of frailty status is in significant correlation with postoperative outcomes (−2 Log (likelihood), df 18, Chi^2^, 34.795, Pr > Chi^2^ = 0.010). The type II analysis table below (Table 3) depicts the contribution and significance of each variable dependent on frailty status, showing a significantly higher probability of PHLF (LR 6.793, *p* = 0.009), bleeding (LR 9.541, *p* = 0.002) and longer ICU stay (LR 8.666, *p* = 0.003) in F patients. The predictions for each patient’s frailty status (NF vs. F) imputed by the logit regression model are represented in the ROC curve (Figure 2) with an AUC of 0.710 (good prediction).

When stratified based on their mFI score (mFI 0 vs. mFI 1 vs. mFI ≥ 2), we found a directly proportional increase in the rate of PHLF (LR 7.730, *p* = 0.005), postoperative bleeding/hematoma (LR 5.052, *p* = 0.025), morbidity (LR 2.761, *p* = 0.09) and major complications (CD 3–4) (LR 2.890, *p* = 0.089) (Table 4); however, in terms of medical complications, HAI, SSI, bile leak, LOS and 30-day mortality, there was no significant increase in frequency along with a higher mFI score.

### 3.4. Index Disease

Based on the hypothesis that frail patients with liver metastases undergoing liver resection may fare worse than patients with primary liver cancer (i.e., hepatocarcinoma, cholangiocarcinoma), another subgroup analysis was performed to assess surgical outcomes of frail patients undergoing liver resection for liver metastases. A total of 106 patients were diagnosed with liver metastases, with significantly more in the NF group (NF, *n* = 57 (63.3%) vs. F, *n* = 49 (35%), *p* < 0.00002). Logistic regression (logit binary model) was performed again, showing (Table 5) that the presence of liver metastasis alone does not correlate with worse surgical outcomes in frail patients. Figure 3 shows the ROC curve of this prediction model with an AUC of 0.708. Figure 4 depicts a comparative predictive value between frail patients with major liver resection vs. minor and patients with primary liver vs. liver metastases. The highest sensitivity and specificity (AUC > 0.7, good predictive value) is seen in patients undergoing major liver resections for liver metastases.

### 3.5. Predicting Posthepatectomy Liver Failure in Frail Patients

Given that frailty is associated with an increased risk of PHLF, a logit regression model was performed to assess which combination of risk factors has the highest predictive value and can be used by clinicians as a prediction normogram. After performing a multivariate analysis of all qualitative and quantitative variables, index disease (primary vs. secondary liver cancer), frailty status, gender and history of viral hepatitis were chosen based on their log-ratios. This normogram was chosen to predict PHLF. The test for the null hypothesis (H0) was significant, with a Chi^2^ of 17.897, *p* = 0.006. Table 6 shows the impact of each variable on the final outcome (PHLF). Figure 5 shows the cumulative predictive value of this normogram illustrated through a ROC curve, with an AUC of 0.913 (excellent).

## 4. Discussion

When considering major liver resections, F patients have a significantly higher probability of developing PHLF and postoperative bleeding and have a longer ICU stay compared to NF patients, and this risk is increased along with higher mFI scores; however, this does not translate into a higher morbidity or mortality rate, nor does it increase the overall length of stay. Even more, when considering all liver resections (minor and major), F patients may have a higher morbidity rate based on more in-hospital infections, but still, they have similar rates of surgical complications, similar rates of PHLF, similar mortality and similar lengths of stay. Noteworthy for clinicians is that frailty is correlated with a higher risk of PHLF after major hepatectomy, especially for an mFI of more than 2, where the rate of PHLF is nearly eight times higher. More so, based on our prediction normogram, male, frail patients undergoing hepatectomy for primary liver cancer having a history of viral hepatitis have the highest risk of PHLF (AUC 0.913).

Our study comes to ignite debate as it contradicts other single-cohort studies, which showed frailty to increase postoperative morbidity, 30-day mortality and the duration of hospitalisation. Yamada et al. [17] carried out a case–control study on 92 patients who underwent minor and major hepatectomies and showed a higher rate of Clavien–Dindo (CD) III complications and a longer length of stay; however, all major surgical complications (i.e., overall morbidity, bile leak, PHLF, blood loss and mortality) were similar between NF and F. More so, the statistical power of CD III complications and length of stay in this study is, at most, borderline significant based on the resulting *p* values. In a cohort of 409 patients, McKechnie et al. [18] showed a longer length of stay and a higher rate of postoperative complications (both minor and major) in F patients after liver resection. In their multivariate analysis, frailty was an independent risk factor for postoperative morbidity. The largest study to date comes from Maegawa et al. [19] conducted on 24.510 patients extracted from the National Surgical Quality Improvement Program (NSQIP) database. Like our study, they found F patients to have a significantly higher incidence of PHLF and more grade IV CD complications, while the rate of bile leak was similar between NF and F.

Predicting outcomes after oncological liver resections is an important topic considering that morbidity in these patients can be as high as 30% [20]. Currently used general scores fail to predict real outcomes in patients. The American College of Surgeons (ACS) National Surgical Quality Improvement Program (NSQIP) score [21] was unable to predict surgical outcomes after liver resections. Frailty is believed to better reflect patients’ biological age and their ability to withstand major surgical stress and could be used alongside other scores to better predict outcomes after hepatectomy, but the current data are scarce and heterogeneous. So far, we do not have a unanimously accepted score of frailty, which makes inter-study comparability difficult. There seems to be a solid body of evidence suggesting a higher risk of PHLF in F patients, and this is important to know for clinicians involved in the prehabilitation of frail patients planned for major liver surgery, as their frail status might be as important as their preoperative liver function quantified through Child–Pugh or MELD scores. Also, the data we have so far seem to agree with the fact that F patients are expected to require longer hospitalisation and, as in our study, may require longer ICU stays. Short-term mortality seems to also be, in other studies, higher in F patients.

There are limitations to our study. First, the retrospective nature of the study implies a high risk of selection, recall and misclassification bias, although the data were extracted by a higher surgical trainee and verified by two senior surgeons. We used the mFI-5 to quantify frailty as it requires less data, and, as such, it might not have been as predictive as other scores, although the literature so far showed mFI-5 to be comparable to mFI-11. The low number of patients, although adequate given the subspecialised field, produced outcomes with a low number of observations, frequently less than five, affecting the statistical power of our analysis. Propensity score matching could not be performed because F patients are by default significantly different in terms of characteristics and comorbidities, meaning that matching to an NF patient was impossible.

## 5. Conclusions

Frailty is associated with a higher probability of developing posthepatectomy liver failure, higher incidence of postoperative bleeding and a longer ICU stay in patients undergoing major liver resection as suggested by other studies; however, these are not increased in patients undergoing minor liver resection, which is new and clinically important as surgeons may perform minor liver resections without an additional perioperative risk. Surgical complications such as bile leak, surgical site infection or 30-day mortality are similar between frail and non-frail patients.

## Figures and Tables

**Figure 1 diagnostics-15-00512-f001:**
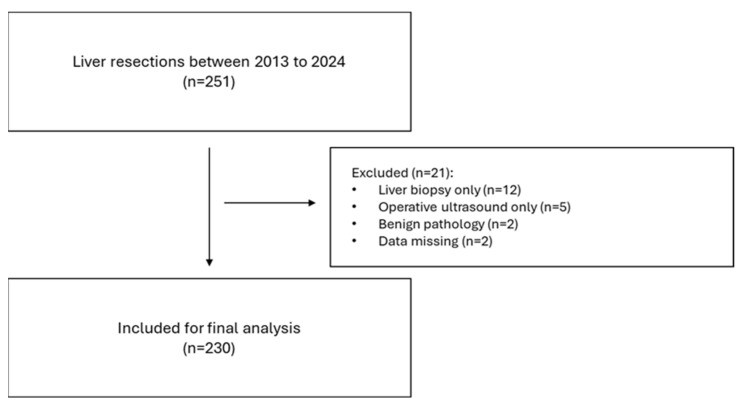
STROBE flowchart.

**Figure 2 diagnostics-15-00512-f002:**
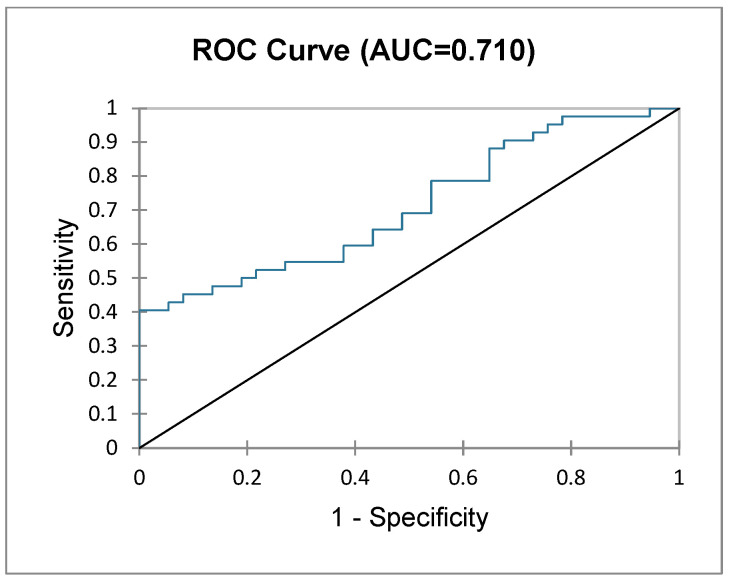
ROC curve depicting frailty status based on postoperative outcomes in patients undergoing major liver resection.

**Figure 3 diagnostics-15-00512-f003:**
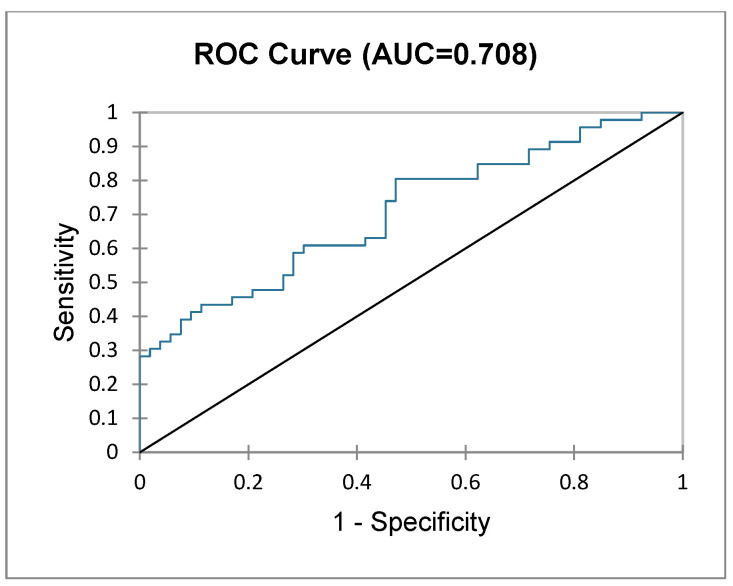
ROC curve depicting frailty status based on postoperative outcomes in patients undergoing surgery for liver metastases.

**Figure 4 diagnostics-15-00512-f004:**
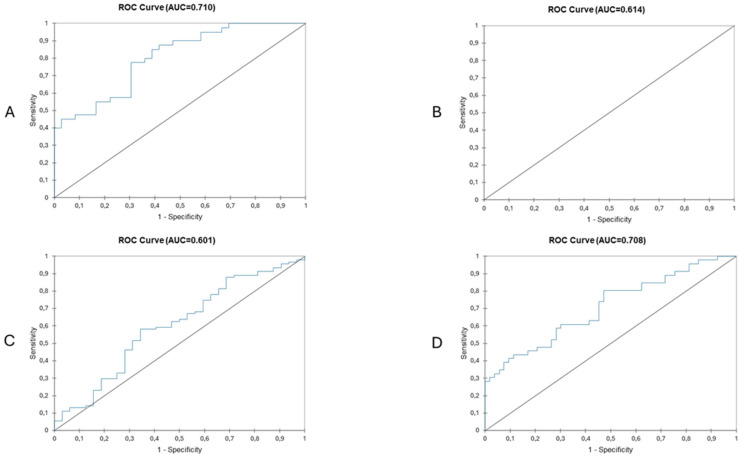
Predictive value of frailty status based on surgical outcomes in patients undergoing major liver resection (**A**), patients undergoing minor liver surgery (**B**), patients with primary liver cancer (**C**) and patients with secondary liver cancer (**D**). Highest AUCs are seen in patients with secondary liver cancer undergoing major liver resection.

**Figure 5 diagnostics-15-00512-f005:**
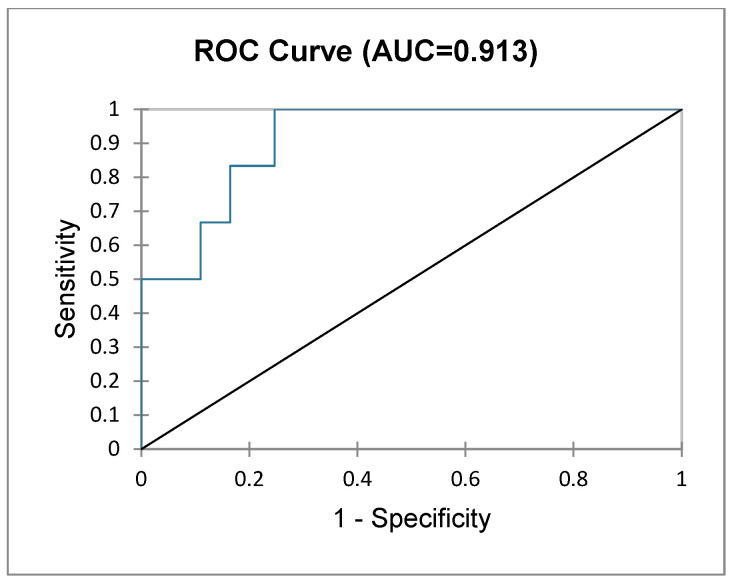
Cumulative sensitivity and specificity of the imputed normogram in predicting PHLF.

**Table 1 diagnostics-15-00512-t001:** Two-group comparison in terms of covariates and preoperative characteristics.

	NF (mFI 0) *n* (%)/Mean (SD)	F (mFI ≥ 1) *n* (%)/Mean (SD)	*p* Value
Total	**90 (100%)**	**140 (100%)**	
Gender (males)	50 (55.5%)	89 (63.5%)	*p* = 0.269
Age, mean (SD)	57.3 (9.6)	66.4 (8.2)	** *p* ** ** < 0.0001 (MD = 1.04)**
BMI	25.0 (4.0)	27.2 (4.3)	** *p* ** ** = 0.0001 (MD = 0.53)**
mFI			
0	90 (100%)	0	
1	0	75 (53.5)	
2	0	46 (32.8)	
3	0	15 (10.7)	
4	0	2 (1.4)	
Comorbidities			
ALD	5 (5.5)	12 (8.5)	*p* = 0.449
Viral hepatitis	17 (18.8)	62 (44.2)	** *p* ** ** = 0.0001**
Cirrhosis	10 (11.1)	47 (33.5)	** *p* ** ** = 0.0001**
Diabetes	3 (3.3)	50 (35.7)	** *p* ** ** < 0.00001**
CHF	0	31 (22.1)	** *p* ** ** < 0.00001**
COPD	1 (1.1)	19 (13.5)	** *p* ** ** = 0.0006**
CKD	4 (4.4)	32 (22.8)	** *p* ** ** = 0.0001**
Smoking	24 (26.6)	43 (30.7)	*p* = 0.554
Neoadjuvant RT	0	5 (3.5)	*p* = 0.159
Neoadjuvant CHT	28 (31.1)	28 (20)	*p* = 0.060
Neoadjuvant RCHT	16 (17.7)	13 (9.2)	*p* = 0.068
ASA 2	30 (33.3)	22 (15.7)	** *p* ** ** = 0.002**
ASA 3	53 (58.8)	102 (72.8)	** *p* ** ** = 0.031**
Preoperative blood tests			
Total bilirubin (mg/dL)	0.8 (0.7)	0.7 (0.3)	*p* = 0.137
Albumin (g/dL)	4.5 (0.5)	4.4 (0.6)	*p* = 0.190
INR	1.05 (0.1)	1.06 (0.1)	*p* = 0.460
PLT (mmc) × 10^3^	239 (110)	201 (82)	** *p* ** ** = 0.003**
AST	37.3 (26.3)	39.2 (22.8)	*p* = 0.562
ALT	36 (29.8)	38.6 (27.4)	*p* = 0.498
Operative factors			
Major hepatectomy	37 (41.1)	42 (30)	*p* = 0.089
No. of liver lesions (≥2)	25 (27.7)	44 (31.4)	*p* = 0.658
Operative time (min)	263 (120)	224 (86)	*p* = 0.004 (MD = 0.39)
Blood loss (mL)	709 (629)	608 (709)	*p* = 0.272

Key: Fisher’s exact test was used, instead of the chi-square test, to compare the number of events between groups, as for some variables, there were less than 5 observations. Two-sample *t*-test (pooled variance) was used to compare means. NF, non-frail; F, frail; SD, standard deviation; MD, mean difference; BMI, body mass index; mFI, modified frailty index; ALD, alcoholic liver disease; CHF, chronic heart failure; COPD, chronic obstructive pulmonary disease; CKD, chronic kidney disease; RT, radiotherapy; CHT, chemotherapy; RCHT, radio-chemotherapy; ASA, American Society of Anesthesiologists Classification; INR, International Normalised Ratio; PLT, platelets; AST, Aspartate Transferase; ALT, Alanine Transaminase.

**Table 2 diagnostics-15-00512-t002:** Postoperative outcomes in NF vs. F patients.

	NF (mFI 0) *n* (%)/Mean (SD)	F (mFI ≥ 1) *n* (%)/Mean (SD)	*p* Value
Total	**90 (100%)**	**140 (100%)**	
Morbidity	12 (13.3)	34 (24.2)	** *p* ** ** = 0.04**
Clavien–Dindo III–V	7 (7.7)	13 (9.28)	*p* = 0.812
Surgical complications			
Bile leak	2 (2.2)	0	*p* = 0.152
PHLF	1 (1.1)	5 (3.5)	*p* = 0.408
Bleeding/hematoma	0	4 (2.8)	*p* = 0.157
IAC	4 (4.4)	6 (4.2)	*p* = 1
SSI	0	2 (1.4)	*p* = 0.521
HAI	6 (6.6)	26 (18.5)	** *p* ** ** = 0.011**
Medical complications			
DVT/PE	2 (2.2)	1 (0.7)	*p* = 0.562
HAP	3 (3.3)	8 (5.7)	*p* = 0.534
Cardiac	2 (2.2)	6 (4.2)	*p* = 0.486
AKI	6 (6.6)	11 (7.8)	*p* = 0.801
MODS	0	5 (3.5)	*p* = 0.159
Reintervention	1 (1.1)	5 (3.5)	*p* = 0.408
LOS	12.8 (5.3)	13.5 (5.7)	*p* = 0.351
Length of ICU stay	2.7 (2.6)	3.3 (4.5)	*p* = 0.252
30-day mortality	1 (1.1)	6 (4.2)	*p* = 0.251

Key: Fisher’s exact test was used, instead of chi-square test, to compare the number of events between groups, as for some variables, there were less than 5 observations. Two-sample *t*-test (pooled variance) was used to compare means. NF, non-frail; F, frail; SD, standard deviation; PHLF, posthepatectomy liver failure; IAC, intraabdominal collection; SSI, surgical site infection; HAI, hospital-acquired infection; DVT, deep vein thrombosis; PE, pulmonary embolism; HAP, hospital-acquired pneumonia; AKI, acute kidney injury; MODS, multiple organ dysfunction syndrome; LOS, length of stay.

**Table 3 diagnostics-15-00512-t003:** Type II analysis table for each of the explanatory variables for frailty status.

Variable	DF	Chi-Square (LR)	Pr > LR
Morbidity	1	0.453	0.501
Clavien–Dindo III–V	1	0.065	0.501
Surgical complications			
Bile leak	1	2.637	0.104
PHLF	1	6.793	**0.009**
Bleeding/hematoma	1	9.541	**0.002**
IAC	1	0.452	0.501
SSI	1	3.788	0.052
HAI	1	2.533	0.112
Medical complications			
DVT/PE	1	3.754	0.053
HAP	1	2.086	0.149
Cardiac	1	0.000	1.000
AKI	1	0.758	0.384
MODS	1	0.457	0.499
Reintervention	2	0.620	0.733
LOS	1	2.323	0.127
Length of ICU stay	1	8.666	**0.003**
30-day mortality	1	0.015	0.904

Key: Type II analysis describes the probabilistic (Pr) influence of each explanatory variable on frailty status based on the Chi^2^ likelihood ratio (LR) and its degree of freedom (DF). PHLF, posthepatectomy liver failure; IAC, intraabdominal collection; SSI, surgical site infection; HAI, hospital-acquired infection; DVT, deep vein thrombosis; PE, pulmonary embolism; HAP, hospital-acquired pneumonia; AKI, acute kidney injury; MODS, multiple organ dysfunction syndrome; LOS, length of stay.

**Table 4 diagnostics-15-00512-t004:** Type II analysis table showing the impact of mFI scores on surgical outcomes.

Variable	mFI = 1		mFI ≥ 2	
	Chi-Square (LR)	Pr > LR	Chi-Square (LR)	Pr > LR
Morbidity	2.103	0.147	2.761	0.097
CD 3–4	0.001	0.982	2.890	0.089
PHLF	5.776	0.016	7.730	0.005
Bleeding	3.797	0.051	5.052	0.025

Key: Type II analysis describes the probabilistic (Pr) influence of each explanatory variable (mFI score) on surgical outcomes based on the Chi^2^ likelihood ratio (LR) and its degree of freedom (DF). CD, Clavien–Dindo; PHLF, posthepatectomy liver failure.

**Table 5 diagnostics-15-00512-t005:** Type II analysis table for each of the explanatory variables for frailty status.

Variable	DF	Chi-Square (LR)	Pr > LR
Morbidity	1	0.296	0.586
Clavien–Dindo III–V	1	0.326	0.568
Surgical complications			
Bile leak	1	1.696	0.193
PHLF	1	0.000	1.000
Bleeding/hematoma	1	3.242	0.072
IAC	1	0.121	0.728
SSI	1	1.512	0.219
HAI	1	0.104	0.747
Medical complications			
DVT/PE	1	1.345	0.561
HAP	1	1.228	0.268
Cardiac	1	0.000	1.000
AKI	1	0.444	0.505
MODS	1	0.000	1.000
Reintervention	2	0.175	0.916
LOS	1	2.684	0.101
Length of ICU stay	1	2.281	0.131
30-day mortality	1	0.000	1.000

Key: Type II analysis describes the probabilistic (Pr) influence of each explanatory variable on frailty status based on the Chi^2^ likelihood ratio (LR) and its degree of freedom (DF). PHLF, posthepatectomy liver failure; IAC, intraabdominal collection; SSI, surgical site infection; HAI, hospital-acquired infection; DVT, deep vein thrombosis; PE, pulmonary embolism; HAP, hospital-acquired pneumonia; AKI, acute kidney injury; MODS, multiple organ dysfunction syndrome; LOS, length of stay.

**Table 6 diagnostics-15-00512-t006:** Type II analysis table for each of the explanatory variables for PHLF.

Variable	DF	Chi-Square (LR)	Pr > LR
Index disease	1	1.216	0.270
Frailty status	3	2.238	0.524
Gender	1	1.815	0.178
Viral hepatitis	1	3.050	0.081

Key: Type II analysis describes the probabilistic (Pr) influence of each explanatory variable on PHLF based on the Chi^2^ likelihood ratio (LR) and its degree of freedom (DF).

## Data Availability

The data that support the findings of this study are available on request from the corresponding author, S.M.

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
