# Peer review of "Is Frailty Associated with Worse Outcomes After Major Liver Surgery? An Observational Case–Control Study"

_diagnostics, 2025, doi:10.3390/diagnostics15050512_

Round 1
Reviewer 1 Report
Comments and Suggestions for Authors
Thank you, I have read the manuscript with interest. Here are some comments:
The paper is not groundbreaking, as it is already well established that frailty is associated with worse outcomes (essentially in any disease).
Nevertheless, to truly understand the impact of frailty, I believe it is important to include the index disease. In geriatrics, frailty is often perceived as a standalone syndrome separate from the index disease. In reality, it is not, but by adjusting for the index disease, at least an attempt can be made to address this. This is especially relevant for this manuscript because primary and secondary tumors are grouped together. I do not see any corrections for this (although there is indirect evidence in Table 1, such as fewer major hepatectomies and more lesions in frail patients). If these are all cancer patients with metastases versus those with primary tumors, it is no surprise that the former group fares worse. This should, in my view, be analyzed separately from frailty.
The abbreviations NF and F add nothing for me.
I expected a Kaplan-Meier curve (or Cox regression or at least logistic regression) stratified by frail vs. non-frail, with adjustments for all relevant factors (those listed in Table 1, but also the index disease or primary vs. secondary tumors).
I am not sure what Figure 2 adds. Idem for ROC-curve (if you don't show KM or PHREG)
Is there no longer follow-up data available?
ABSTRACT: Please avoid relying solely on p-values; include the actual numerical data. If additional analyses (as mentioned above) are conducted, numerical values such as OR, HR or survival rates should be provided, depending on the type of analysis performed
Author Response
Dear Reviewer,
Thank you for giving us the opportunity to submit a revised manuscript in this journal. We appreciate the time and effort you dedicated to our study, and we are grateful for the interesting comments and valuable suggestions you have made to the paper.
In this revised manuscript we have incorporated most of the comments and highlighted the changes to the text in red. Please find bellow our point-by-point reply.
Comment 1: The paper is not groundbreaking, as it is already well established that frailty is associated with worse outcomes (essentially in any disease).
Answer: Thank you for this remark. Indeed, frailty has been associated with worse outcomes in various diseases, the literature being abundant on this subject, however, so far frailty failed to be implemented in standard clinical practice in the preassessment of surgical candidates. With regards to liver surgery, the data we have so far is conflicting. There are studies that show higher surgical and medical morbidity, higher mortality, longer length of stay and wore overall survival in frail patients, however others failed to prove that frailty can predict outcomes (Milliken D, Curtis S, Melikian C. Predicting morbidity in liver resection surgery: external validation of the revised frailty index and development of a novel predictive model. HPB (Oxford). 2021 Jun;23(6):954-961. doi: 10.1016/j.hpb.2020.10.012. Epub 2020 Nov 6. PMID: 33168438.). More so, we believe that the results should be stratified based on extent of resection (minor vs major). Our hypothesis, derived from our own experience, was that outcomes should be similar when only minor metastasectomies are performed, whereas for major resection, frailty may indeed have an impact and thus frail patients proposed for a major resection should be better informed about their risks. Similarly, even frail, elderly patients that require a minor liver resection should be aware that postoperative results are similar to younger, non-frail counterparts, which is clinically important. Our study delivers on this issue, being probably the only comparative study which stratifies patients in this manner. More so, while many studies compared overall morbidity, they did not report the rate of postoperative hepatic failure (PHLF), which is the most notorious complication after a major liver resection. Our study focused on finding whether PHLF is higher in frail patients, and indeed, it is. This is clinically important for liver surgeons when deciding to perform a major liver resection in a frail patient.
Comment 2: To truly understand the impact of frailty, I believe it is important to include the index disease. In geriatrics, frailty is often perceived as a standalone syndrome separate from the index disease. In reality, it is not, but by adjusting for the index disease, at least an attempt can be made to address this. This is especially relevant for this manuscript because primary and secondary tumors are grouped together. I do not see any corrections for this (although there is indirect evidence in Table 1, such as fewer major hepatectomies and more lesions in frail patients). If these are all cancer patients with metastases versus those with primary tumors, it is no surprise that the former group fares worse. This should, in my view, be analyzed separately from frailty.
Answer: Thank you for this suggestion. We agree that a subgroup analysis should`ve been performed based on patients index disease. In the revised version we have added a new subchapter (see subchapters 3.4 and 3.5, Lines 175-222) in results and assessed outcomes in frail patients undergoing surgery for liver metastases, however we could not find a higher risk of complications in this group. We went further to see whether index disease might influence the risk of posthepatectomy liver failure (which is our primary outcome in this study) and found that patients with primary liver cancer have a higher risk of PHLF. For this reason, we have performed a multivariate logistic regression and found that the variables which influence the risk of PHLF the most are index disease, frailty status, gender and history of viral hepatitis. We have tested this normogram and, indeed, in this category of patients (male, frail, primary liver cancer and viral hepatitis) the AUC is 0.912, excellent predictive value for PHLF.
Comment 3: The abbreviations NF and F add nothing for me.
Answer: We are sorry if this abbreviation is redundant, but we would use the terms frail and non-frail very often in the study, thus it helps in reducing repetition, save space and summarizing info, especially in tables. To make it easier we have defined the terms F and NF in each table and in the text.
Comment 4: I expected a Kaplan-Meier curve (or Cox regression or at least logistic regression) stratified by frail vs. non-frail, with adjustments for all relevant factors (those listed in Table 1, but also the index disease or primary vs. secondary tumors).
Answer: Thank you for your suggestion. We have performed binary logistic regression between frail and non-frail as a whole but also stratified based on type of surgery (major vs minor) and based on index disease as you kindly suggested. Unfortunately, we could not perform Kaplan-Meier or Cox proportional hazards model because we don`t have long term follow-up as it was not the aim of this study and both analyses require time to event data.
Comment 5: I am not sure what Figure 2 adds. Idem for ROC-curve (if you don't show KM or PHREG)
Answer: Thank you for this remark. We have deleted the confusion plot as you suggested. Indeed, it is not widely used in literature for depicting the performance of a prediction model. However, we kept and believe the ROC-curves are welcomed for emphasizing the predictive value of our imputed algorithm. Even without time-to-event data (as we would need in Kaplan-Meier or proportional hazards), ROC-curves can be used to represent the sensitivity and specificity of a certain variable or group of variables (as in our case) in predicting a certain outcome (e.g., posthepatectomy liver failure). More so, it gives the prediction a clear numerical value through the AUC helping readers to easily understand the significance of the correlation.
Comment 6: Is there no longer follow-up data available?
Answer: Thank you. Unfortunately, we do not have long term follow-up data. The aim of this study was to assess short term outcomes between frail and non-frail undergoing oncological liver resections. Because frail patients are naturally older and have more comorbidities than their counterparts, it is likely that their overall survival will be worse, but that doesn`t really help surgeons or clinicians involved in the prehabilitation and decision making of surgical candidates. What we aimed to understand is whether frailty does increase morbidity and mortality, to what extent and whether we should consider less extensive surgery in this category (parenchymal sparring rather than major hepatectomy). To answer this, short term outcomes are sufficient because these will be affected by the operation more than their survival which could be the results of other preexisting conditions.
Comment 7: ABSTRACT: Please avoid relying solely on p-values; include the actual numerical data. If additional analyses (as mentioned above) are conducted, numerical values such as OR, HR or survival rates should be provided, depending on the type of analysis performed
Answer: Thank you. We have amended this.
Reviewer 2 Report
Comments and Suggestions for Authors
The authors investigated 230 patients qualified for liver resection due to cancer in order to find out if frailty existing prior to the surgical treatment constitutes a significant factor related to further outcomes. The survey raises an essential clinical problem. I have several concerns:
-
The manuscript doesn’t include the scheme of 5-item modified frailty index (mFI-5) - assessed parameters aren’t presented. Please, include it.
-
Did the authors try to categorize patients in a more detailed way according to frailty index: results not only of 1 or more points as a single parameter, but: 1-2-3-4 as independent categories - are any significant differences comparing these subgroups between each other?
-
The authors admitted that similar results were achieved in other previously performed investigations. Therefore, please explain, why did you decide to follow this path and what is the novelty or significant contribution of this paper in this field of medicine. It should be included within the section of conclusions.
Author Response
Dear Reviewer,
Thank you for giving us the opportunity to submit a revised manuscript in this journal. We appreciate the time and effort you dedicated to our study, and we are grateful for the interesting comments and valuable suggestions you have made to the paper.
In this revised manuscript we have incorporated most of the comments and highlighted the changes to the text in red. Please find bellow our point-by-point reply.
Comment 1: The manuscript doesn’t include the scheme of 5-item modified frailty index (mFI-5) - assessed parameters aren’t presented. Please, include it.
Answer: Thank you for this remark. We have added a paragraph defining the mFI-5 and its parameters (Lines 79-82).
Comment 2: Did the authors try to categorize patients in a more detailed way according to frailty index: results not only of 1 or more points as a single parameter, but: 1-2-3-4 as independent categories - are any significant differences comparing these subgroups between each other?
Answer: Thank you for this fine suggestion. In the revised manuscript we have split patients into mFI=1 and mFI more than 2, performed a multivariate logistic regression and found that the risk of PHLF, bleeding, morbidity and major complications (Clavien-Dindo 3-4) increases directly proportional with higher mFI scores. This is clinically important, as patients with mFI≥2 have a eight times higher rate o PHLF. We also included a new table describing the data for the stratified mFI`s (Lines 177-185) and we thank you for it.
Comment 3: The authors admitted that similar results were achieved in other previously performed investigations. Therefore, please explain, why did you decide to follow this path and what is the novelty or significant contribution of this paper in this field of medicine. It should be included within the section of conclusions.
Answer: Thank you for this remark. Our results are partially in agreement with other studies as mFI can predict surgical outcomes, but contrary to current literature, we found that worse outcomes are seen only after major liver resections, whereas after minor liver resections (less than 3 segments), frailty alone does not seem to affect morbidity or 30-day mortality. Also, our study brings new insight on the risk of posthepatectomy liver failure, which is the most notorious surgical complication after a major liver resection. In the revised version, in the conclusions section, we have highlighted how our study is in agreement but, in the same time, necessary in addition to other already published studies.
Round 2
Reviewer 1 Report
Comments and Suggestions for Authors
Thank you for the revisions. I believe we still have differing opinions on the potential analyses that could be conducted with this data. That being said, it is not my role as a peer reviewer to convince the authors otherwise. I can accept this revision, which I see as a reasonable compromise, where the authors now also appropriately discuss the index disease (and addressed other relevant issues)
Thank you!
Reviewer 2 Report
Comments and Suggestions for Authors
Thank you for the introduced improvements.